# Mexican Validation of the Engagement and Disaffection in Physical Education Scale

**DOI:** 10.3390/ijerph17061821

**Published:** 2020-03-11

**Authors:** René Rodríguez-Medellín, Jorge Zamarripa, María Marentes-Castillo, Fernando Otero-Saborido, Raúl Baños, Raquel Morquecho-Sánchez

**Affiliations:** 1Autonomous University of Nuevo Leon, Faculty of Sports Organization. Cd. Universitaria, s/n, San Nicolás de los Garza, Nuevo Leon 66451, Mexico; 2Pablo de Olavide University, Faculty of Sports Sciences. Ctra. de Utrera, 1, Sevilla 41013, Spain; 3Autonomous University of Baja California, Faculty of Sports, Campus Ensenada. Paseo de la Playa, Carlos Pacheco 7, Ensenada 22890, Mexico

**Keywords:** engagement, disaffection, physical education, invariance, gender, Mexico

## Abstract

To date, no instrument adapted and validated that measures engagement and disaffection in the physical education class has been found, which limits the generation of knowledge of this area in Mexico. The aims of this study were to translate and adapt the engagement and disaffection scale to the context of physical education in Mexico and to examine its reliability, structure (two and four factors), and factorial invariance by gender in Mexican fifth- and sixth-grade elementary school students. A total of 1470 students participated (50.6% boys) with ages between 10 and 14 years (mean (*M*) = 10.56; standard deviation (*SD*) = 0.77) from federal (89.3%) and state (10.7%) elementary schools. Two factorial structures were tested (with four factors and two factors). The fit indexes of both models were satisfactory, and the factorial saturations were significant. The differences between the fit indexes of both models were irrelevant; therefore, the two-factor model was considered more suitable. The total strict invariance by gender was confirmed, and the reliabilities of the engagement and disaffection scale were acceptable. The Mexican version of the course engagement and disaffection scale in physical education is valid and useful to measure these constructs in the context of physical education in Mexico.

## 1. Introduction

Within the education context, engagement is seen as a malleable state that is influenced by different processes, like the school’s, teacher’s, parent’s, and classmate´s ability to provide constant support to achieve learning [1,2,3,4]. It is an active image that represents learning through the effort and interaction with the teacher; in other words, it is both an individual and a context issue.

Research regarding engagement in the school context has a background and consequences similar to those found in the labor context, such as self-efficacy, autonomy, social support [5], optimism, and hope [6]. In addition, it is believed that it is especially important for apathetic and discouraged students and those with a high risk of abandoning [7]; therefore, it is an elemental concept to understand the phenomenon of desertion, as well as to promote a successful educational path [8].

These findings highlight the importance of examining academic engagement as an indicator of wellbeing in student populations, as well as a motivational agent for promoting positive consequences, such as performance and learning [9]. However, in the understanding of this engagement construct, there are two important areas that generate confusion. The first focuses on the lack of a clear distinction between indicators (characteristics that belong to engagement itself) versus facilitators (factors outside of the construct that are thought to influence engagement), and the second refers to the number and nature of the dimensions within the engagement; in other words, how many should be distinguished and if they have internal dynamics.

In the scientific literature, there are different theoretical approaches and conceptualizations regarding engagement [10]. Some state that it is a meta-construct that is composed of multiple dimensions on the participation in school [11]. Others emphasize the mental state related to the study of activities [5]. Both models focus on the components of engagement. However, the model by Fredricks, Blumenfeld, and Paris [11], also called the North American model, proposes that engagement is composed of three dimensions: behavior, emotion, and cognitive. On the other hand, the model by Schaufeli, Salanova, Gonzalez-Romá, and Bakker [5], also known as the European concept of engagement, proposes that three dimensions characterize engagement: absorption, vigor, and dedication.

Skinner, Furrer, Marchand, and Kindermann [12] proposed a bipartite model of engagement (E) composed of two dimensions: emotional and behavioral, which are closely linked in the classroom; that is, they are inter-individually stable and are formed in the same way by external factors without influencing each other. Theories such as the Self-Determination Theory (SDT) [13,14] support this conceptualization by suggesting that emotions feed behaviors in the classroom; in other words, emotional engagement, such as interest and enthusiasm, feed behavioral engagement, such as effort and persistence.

Emotional engagement (EE) focuses on the states that are relevant to the emotional participation of the students during learning activities, such as enthusiasm, interest, and enjoyment. On the other hand, behavioral engagement (BE) includes the effort, attention, and persistence during the initiation and execution of learning activities [15]. Engagement in itself combines both dimensions and refers to active interactions, oriented towards objectives: flexible, constructive, persistent, focused, and emotionally positive, with the social and physical environments. In this case, the activities of the physical education (PE) class. Therefore, academic engagement refers to what you want to generate in students through the school context.

In this study, the conceptualization of engagement used by other authors [12,16,17,18] will be used, as well as the approach of negative engagement referred to in the literature as disaffection [19,20].

Disaffection (D) occupies the negative pole of the engagement continuum and refers to the occurrence of behaviors and emotions that reflect motivational states of poor adaptation [18].

Like engagement, disaffection is composed of the emotional dimension (ED), which includes boredom, anxiety, and frustration in the classroom, and the behavioral dimension (BD), which includes passivity and withdrawal from participation in learning activities [12]. This is because alienation or disaffection probably reflects more than a lack of participation [21]; in other words, when the person does not have the option of abandoning an activity, as normally occurs in school, mental or emotional disaffection can occur [20]. Thus, disaffection is an important source of motivational impotency in children, which impedes achievement in PE [18].

The results of studies in the context of PE have demonstrated the relationship between the student’s engagement and academic performance [22], the psychological needs of autonomy, competence, and relationships [23], which, from SDT, are considered as basic and universal for optimal development and growth, as well as for the wellbeing of individuals [24]. Other studies point out the influence of the teacher figure on student engagement. The teacher´s behavior [23] and his or her relationship with their students [25] promote engagement during PE class. Also, extracurricular physical activity is another variable that is directly influenced by engagement and by the perception of student competence in the PE class [26].

These studies demonstrate the importance of measuring engagement as an integral part of any education system since the results obtained from this type of study will serve as very useful indicators in classrooms and be of great value for future scientific research, as well as for the early detection of disinterest and educational abandonment, which will be the objectives of interventions aimed at increasing students’ engagement to school and learning.

To date, no instrument adapted and validated to the Mexican context that measures engagement, disaffection, and their respective dimensions in the PE class has been found. This has limited the generation of knowledge of this area in Mexico, and, given the positive relationship that exists between engagement with the PE class and the amount of physical activity that is carried out outside of school [26], studies in this area could contribute to reducing the high rates of physical inactivity that exist in the Mexican population from an early age [27]. Therefore, the objective of this study is to translate the School Engagement Scale of Chi, Skinner, and Kindermann [28] into Spanish and adapt it to the context of PE in Mexico, examining its psychometric properties, structure, and factorial invariance by gender in a sample of Mexican fifth- and sixth-grade elementary school students from the metropolitan area of Monterrey, Nuevo Leon Mexico.

## 2. Materials and Methods

### 2.1. Study Design and Sample Description

This was a cross-sectional, descriptive, correlational study. This study involved 1470 students (boys = 50.6% and girls = 49.4%) from fifth (49.3%) and sixth (50.7%) grade from 46 public elementary schools (federal = 89.3% and state = 10.7%) in the metropolitan area of Monterrey, Nuevo León, Mexico, with ages from 10 to 14 years (mean age (*M*_age_) = 10.56; standard deviation (*SD*) = 0.77; median = 11) who attended PE class twice a week with a duration of 50 minutes per session, and in which 68% said they practiced at least one sport outside of school. To select the participants, a convenience sample was used considering both gender and grade. Fifth- and sixth-grade students were chosen because children who belong to the final stage of childhood and early adolescence are at the highest level of cognitive development and will not have any complications when responding to the instruments [29].

### 2.2. Instrument

To measure student engagement and disaffection, the Course Engagement and Disaffection Scale (CEDS) [28] was translated and adapted to the context of PE in Mexico. The scale is composed of 12 items grouped into four dimensions: behavioral engagement (BE), emotional engagement (EE), behavioral disaffection (BD), and emotional disaffection (ED). Each one of these indicators was measured by 3 items. The instrument has as a heading “On a Likert scale from 1 (False) to 5 (True), tell us how true each of the following statements is in reference to the physical education classes” (“*En una escala del 1* (*Falso*) *al 5* (*Cierto*), *dinos qué tan ciertas son las siguientes afirmaciones referentes a las clases de educación física*”). One example of the BE is “I pay attention in the physical education class” (“*Pongo atención en la clase de educación física*”) and of the EE, “I enjoy the time I spend in the physical education class” (“*Disfruto el tiempo que paso en la clase de educación física*”). On the other hand, one example of BD is “I only do enough to pass the physical education class” (“*Sólo hago lo suficiente para pasar en la clase de educación física*”), and of ED “The classes of the physical education teacher are very boring” (“*Son muy aburridas las clases del profesor de educación física*”). These items can be grouped in a broader sense where the average of the BE and the EE form the engagement factor (E); on the other hand, the average of the items of the BD and ED form the disaffection factor (D).

### 2.3. Procedure

This study was carried out according to the ethical guidelines recommended by the American Psychological Association (APA). Authorization was requested in writing from the school zone authorities and from each of the principals of the schools explaining the objectives of the research and the procedure that would be performed together with a model of the instrument. Afterward, authorization was requested for application from the teachers of each group and from the selected students taking into consideration the inclusion criteria: be a regular student in their respective group, regularly have PE class at least twice a week, be voluntarily willing to complete the questionnaire, and deliver the informed consent to participate in the research signed by their parents or tutors. The students were informed of the objective of the study, their willingness to volunteer, the absolute confidentiality of their answers, and the management of the data. They were also told that there were no correct or incorrect answers and they were asked for maximum sincerity and honesty. The questionnaire was anonymous and self-administered collectively in the classroom during school hours. To homogenize the data collection conditions, the administrators received prior preparation and training. The protocol was approved by the Ethics Committee of the Autonomous University of Nuevo Leon (No. 16CI19039021). All subjects gave written informed consent in accordance with the Declaration of Helsinki.

The CEDS was translated into Mexican Spanish following the translation–back translation procedure [30]. The translation was carried out by a professional translation agency hired by the researchers. To adapt the translation to the context of PE, a group of experts was formed with two PhD specialists with previous experience in the validation of psychological instruments, a physical education teacher, and a translator specialized in the area of physical activity and sports, who discussed the discrepancies of the translation until the first version of the Mexican Spanish-language instrument was achieved. This version was retranslated into English by a professional translation agency different from the first, and both versions of the instrument were compared: the original and the translation. The differences in the versions were analyzed again and necessary changes were introduced to facilitate comprehension of the items achieving a final version of each of the scales. This version was administered as a pilot application to a group of 72 students (51.40% boys and 48.60% girls; *M*_age_ = 10.56; *SD* = 0.78; range = 10–13) of fifth (54.2%) and sixth grade (45.8%) of an elementary school that was not part of the final sample to verify comprehension of each of the items and define the final version. The selection procedure was the same as described in the section of participants and the results of this pilot application did not show any comprehension problems.

### 2.4. Data Analysis

First, a descriptive analysis was performed for the entire scale and the factors that comprise it. Missing data rates were very small (0.14%) that it was not considered necessary to impute the data. To test the factorial structure of the questionnaire, a confirmatory factor analysis (CFA) was performed of the two proposed models (of two and four factors). Considering the number of response categories of the observable variables (k ≥ 5) and the values range of skewness and kurtosis (see Table 1), the CFA was performed with the maximum likelihood method and as input, and the polychoric correlation and asymptotic covariance matrix were used.

Model adequacy was analyzed with different fit indexes, such as the Comparative Fit Index (CFI), Non-Normed Fit Index (NNFI), and Root Mean Square Error of Approximation (RMSEA). CFI and NNFI values greater or equal to 0.95 indicate an acceptable fit [31]. For RMSEA, negative values or values equal to or lower than 0.08 are considered satisfactory [32]. The evaluation to determine which of the two models (two and four factors) was a better fit for the values, as well as the factorial invariance by gender, was performed using the differences between the goodness-of-fit indexes of the models. It is assumed that there are irrelevant differences between the models and the factorial invariance between groups if ΔCFI and ΔNNFI ≤ 0.01 [33] and ΔRMSEA ≤ 0.015 [34].

The internal consistency of the instrument and the subscales that compose it were assessed using Cronbach’s alpha [35], composite reliability (CR), and the average variance extracted (AVE), as well as a correlation analysis between the factors. The alpha, CR ≥ 0.70, and AVE ≥ 0.50 values are considered acceptable [36]. Convergent validity was analyzed considering that the items had a high burden in their respective construct and the AVE values were ≥ 0.50. Discriminatory validity was examined confirming that the AVE of each construct was superior to the squared correlation between the constructs [36]. The analyses were carried out using the Statistical Package for the Social Sciences (SPSS) V.23 (IBM, Armonk, NY, USA) and the Linear Structural Relations (LISREL) V. 8.80 software [37].

## 3. Results

### 3.1. Descriptive Analysis and Normality

The descriptive analysis (mean, standard deviation, asymmetry, and kurtosis) of each of the items, variables, and factors that composed the scale are shown in Table 1. The results reveal higher engagement than disaffection values with PE. Specifically, emotional engagement had higher values in comparison with behavioral engagement, and in the case of emotional and behavioral disaffection, both had the same mean. Most of the asymmetry and kurtosis values were outside the range (−1.5, 1.5), indicating a normal distribution of data [38].

### 3.2. Confirmatory Factor Analysis (CFA)

The goodness-of-fit indexes of the two-factor, (Satorra-Bentler (SB)χ^2^ = 367.58; degree freedom (*df)* = 52; *p* < 0.01; NNFI = 0.971; CFI = 0.977; RMSEA = 0.064) and of the four-factor model (SBχ^2^ = 239.34; *df* = 48; *p* < 0.01; NNFI = 0.981; CFI = 0.986; RMSEA = 0.052) were satisfactory. All of the factorial saturations of the two models were statistically significant (*p* < 0.05).

The differences between the fit indexes of the two models were irrelevant (ΔNNFI = 0.010; ΔCFI = 0.009; ΔRMSEA = 0.012), both models fit similarly, so these results provide support to the most parsimonious model, that is, the two-factor. In addition, the correlation values between the dimensions EE and BE (*r* = 0.89) and between ED and BD (*r* = 0.80) in the phi matrix of the CFA were high. This suggests that each dimension group formed only one construct; therefore, the two-factor model was the most adequate.

### 3.3. Factorial Invariance by Gender

Taking into consideration the results of the previous section, we proceeded to evaluate the structure invariance of the two factors based on gender. Considering the normal distribution of data [38] (Table 1), maximum likelihood was used as an estimation method and covariance matrices, the mean vector, and the asymptotic covariance matrix were used as input for the multi-sample CFA. First, the structure of the course engagement and disaffection scale in physical education (CEDS-PE) was analyzed separately in the sample of boys (Model M0a) and girls (Model M0b). As shown in Table 2, the goodness-of-fit indexes of the models M0a and M0b were satisfactory and all the estimated parameters were statistically significant (*p* < 0.01).

Later, multi-sample analyses were performed creating new nested models. Model (M1) examined the structural invariance in the two groups showing satisfactory fit indexes, which revealed that the factorial structure of the CEDS-PE is invariant between the two groups.

Model 2 (M2) tested the equivalence of the matrix of the factorial saturations through the boys’ and girls’ group. The goodness-of-fit indexes obtained were satisfactory and the difference obtained between M2 and M1 did not surpass the criterion values; therefore, the invariance in the factorial saturations of the instrument in both samples was confirmed.

Model 3 (M3), which adds the equivalence of the intercepts, showed satisfactory goodness-of-fit indexes. The differences between the goodness-of-fit indexes in the M3 and M1 models did not surpass the criterion values; thus, the equivalence of the factorial saturations and the intercepts was accepted.

Model 4 (M4) added the invariance of the factorial saturations, intercepts, and errors. The results also showed satisfactory goodness-of-fit indexes, and the difference between M4 and M1 did not surpass the criterion values; thus, these results support the strict factorial invariance of the CEDS-PE through gender.

### 3.4. Internal Consistency, Correlations, Convergent and Discriminant Validity

The results of the reliability of the instrument are presented in Table 3. The values of Cronbach’s alpha, CR, and the AVE are acceptable except the AVE of the variable engagement. In general, these results provide support to the convergent validity of the CEDS-PE. On the other hand, the value of the average variance extracted of engagement and disaffection was greater than the squared correlation between both constructs; therefore, these results support the discriminant validity of the CEDS-PE.

## 4. Discussion

In recent years, there has been a growing interest in the school engagement since it has been found that this construct can work as a solution for low academic performance, high levels of boredom and disaffection, and high rates of school dropouts in urban areas [39].

Nevertheless, studies regarding this topic in the Mexican population are still scarce. This could be due to the lack of instruments adapted and validated to the cultural and linguistic context of Mexico; therefore, the aims of this study were to translate the School Engagement Scale of Chi et al. [28] into Mexican Spanish and adapt it to the context of PE, and examine its psychometric properties, structure, and factorial invariance by gender in a sample of Mexican fifth- and sixth-grade elementary school students.

Although engagement is relatively diverse, and researchers have consistently disagreed on the types and number of the dimensions of engagement [11,40,41,42], it seems that a consensus has been reached that the construct is multidimensional and encompasses different aspects. In the present study, the factorial structure of the Mexican version of the CEDS-PE was evaluated by comparing two factorial models, a two-factor model (engagement and disaffection) and another model composed of four indicators (emotional engagement, behavioral engagement, emotional disaffection, and behavioral disaffection).

Results show that both models presented adequate fit of the data; that is, the instrument can be used to measure engagement versus disaffection or with the behavioral and emotional indicators of each. These results are similar to those of Skinner, Furrer, Marchand, and Kindermann [12] and Skinner, Kindermann, and Furrer [20] in the academic domain with children of fourth to seventh grade in a rural-suburban school of New York. On the other hand, these findings contrast with the Immekus and Ingle [43] findings, which obtained a poor data fit of the two- and four-factor model; however, this study was carried out about the implementation of a project based on English language learning, unlike our study that was conducted for PE class.

With respect to the evaluation to determine which of the two models (two and four factors) was a better fit for the values, results show that differences between the fit indexes of the two models were irrelevant so these results provide support to the most parsimonious model, that is, the two-factor. In addition, the high correlations found in the present study between the behavioral and emotional indicators suggest uniqueness. For this two-factor model, different studies have been successfully conducted in different contexts, like academic [44,45] and PE class [46]. However, these results differ from other studies [12,20,28,47], which support the four-dimension model, since it presented the best fit indexes and moderate correlation values between the behavioral and emotional indicators, that is, in the results of these works, the factors are related but distinguishable from each other.

One of the greatest contributions of the present work was to examine the factorial invariance by gender, which had not been considered in previous studies. Considering the aforementioned results, the model that was tested was the two-factor (engagement versus disaffection). The results of the multi-sample CFA supported the strict factorial invariance through gender; therefore, the CEDS-PE is an instrument that can be used to measure the engagement and disaffection of students towards PE class and to perform comparisons between groups of boys and girls.

The analysis of its internal consistency revealed alpha coefficients that meet the acceptable value of 0.70, recommended by Nunnally and Bernstein [48] and are similar to those obtained in other works [12,20,28,44,45]. In addition, the CR and AVE values of disaffection were above the minimum acceptable criterion and the squared correlation between factors [36]. This supports the convergent and discriminant validity of the two-factor structure of the CEDS-PE (engagement versus disaffection).

This study also has some limitations. This study only includes students from elementary schools in the metropolitan area of Monterrey; therefore, future research to analyze the psychometric properties of the instrument with a population from different school levels and sectors of the country should be carried out. This study presents psychometric support of the Spanish version of the instrument in the linguistic and cultural context of Mexico; thus, the study of psychometric properties with populations from other Spanish-speaking countries could be expanded. It is suggested that studies including the factorial invariance according to grades and school levels, areas and populations of other sectors of the country, as well as populations from different Spanish-speaking countries are performed to determine its function and facilitate the comparison of results. Lastly, we suggested studies that examine the effect of teaching practice, the relationship with peers, parental support, and the value and usefulness given to PE on engagement and disaffection.

## 5. Conclusions

The results support the two-factor structure (engagement versus disaffection) and the factorial invariance by gender of the Mexican version of the Course Engagement and Disaffection Scale in Physical Education (CEDS-PE), which is a reliable and valid instrument that can be used by teachers, school principals, institutions responsible for education, and researchers to conduct studies to know the levels of engagement and disaffection of students during PE class and make comparisons between boys and girls. In this way, the present study contributes to the generation of knowledge and scientific production in this area in Mexico.

## Figures and Tables

**Table 1 ijerph-17-01821-t001:** The standardized solution of the four sub-scales of the instrument.

Sub-scales		*M*	*SD*	*Asymmetry*	*Kurtosis*	Factorial saturations	
						*2 factors*	*4 factors*
*Engagement*		4.10	0.72	−1.10	1.49		
	*Emotional engagement*	4.21	0.85	−1.26	1.33		
1	I pay attention in my physical education class (*Pongo atención en la clase de educación física*)	4.38	1.01	−1.81	2.76	0.64	0.67
2	I study for my physical education class (*Estudio para la clase de educación física*)	3.25	1.51	−0.35	−1.32	0.24	0.26
3	I try to do the most I can in the physical education class (*Trato de hacer lo más que pueda en la clase de educación física*)	4.35	0.98	−1.62	2.12	0.74	0.80
	*Behavioral engagement*	3.99	0.82	−0.78	0.49		
4	I enjoy the time I spend in the physical education class (*Disfruto del tiempo que paso en la clase de educación física*)	4.42	1.00	−1.90	3.01	0.77	0.79
5	It is exciting when I make connections between ideas learned in the physical education class (*Es emocionante cuando hago conexiones entre las ideas aprendidas en la clase de educación física*)	4.09	1.14	−1.22	0.72	0.64	0.64
6	The content we see in the physical education class is interesting (*Es interesante el contenido que vemos en la clase de educación física*)	4.12	1.16	−1.28	0.74	0.69	0.69
*Disaffection*		2.21	1.05	0.72	−0.34		
	*Emotional disaffection*	1.93	1.20	1.09	−0.07		
7	It is difficult to attend the physical education class (*Es difícil asistir a la clase de educación física*)	2.25	1.50	0.72	−1.04	0.61	0.71
8	I only do enough to pass the physical education class (*Sólo hago lo suficiente para pasar en la clase de educación física*)	2.85	1.60	0.09	−1.56	0.44	0.59
9	I do not do much work outside the physical education class (*No hago mucho trabajo fuera de la clase de educación física*)	2.35	1.48	0.60	−1.12	0.66	0.78
	*Behavioral disaffection*	1.93	1.20	1.09	−0.07		
10	The classes of the physical education teacher are very boring (*Son muy aburridas las clases del profesor de educación física*)	1.97	1.37	1.10	−0.23	0.86	0.86
11	The physical education class stresses me (*Me estresa la clase de educación física*)	2.00	1.41	1.09	−0.33	0.86	0.86
12	Being in the physical education class is a waste of time (*Es una pérdida de tiempo estar en la clase de educación física*)	1.81	1.34	1.43	0.57	0.90	0.91

*Note: M:* Mean; *SD*: Standard Deviation. All saturations were significant, *t* > 1.96, *p* < 0.05.

**Table 2 ijerph-17-01821-t002:** Goodness-of-fit indexes of the invariance models.

Model description		*df*	SBχ^2^	RMSEA	(90% CI)	NNFI	CFI	ΔNNFI	ΔCFI	ΔRMSEA
M0a	Baseline Model boy	53	245.547 **	0.070	(0.061–0.079)	0.964	0.971			
M0b	Baseline Model girl	53	199.99 **	0.062	(0.053–0.071)	0.974	0.980			
M1	Structural invariance (Baseline Model)	106	444.020 **	0.066	(0.060–0.072)	0.969	0.975			
M2	FL invariance	116	476.339 **	0.065	(0.059–0.071)	0.970	0.974	0.001	0.001	0.001
M3	FL invariance + Int.	126	498.811 **	0.063	(0.058–0.069)	0.972	0.973	0.003	0.002	0.003
M4	FS Invariance + Int. + Error	138	486.970 **	0.059	(0.053–0.064)	0.976	0.975	0.007	0.000	0.007

*Note: df* = degree of freedom; RMSEA = Root Mean Square Error of Approximation; 90% CI = 90% confidence interval for the RMSEA; NNFI = Non-Normed Fit Index; CFI = Comparative Fit Index; FL = factor load; Int.= intercepts. All comparisons in the Δ indices are made with respect to the baseline model (M1); ** *p* < 0.01.

**Table 3 ijerph-17-01821-t003:** Reliability, bivariate correlations, and discriminant validity between the variables of the study.

Dimensions	α	CR	AVE	1	2
1. Engagement	0.70	0.80	0.42	1	0.28
2. Disaffection	0.82	0.87	0.55	−0.53 **	1

*Note:* ** *p* < 0.01; α = Cronbach’s alpha*;* CR = Composite reliability; AVE = Average variance extracted. The value below the diagonal corresponds to the correlation between the variables. The value above the diagonal corresponds to the squared correlation between the variables.

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
