# Peer review of "Mexican Validation of the Engagement and Disaffection in Physical Education Scale"

_ijerph, 2020, doi:10.3390/ijerph17061821_

Round 1

Reviewer 1 Report

The paper seems to be interesting, but in regards of Mexican population (locally). Nevertheless as authors claims there is the  first study undertaken in Mexico in that context .

Introduction

The expression ‘physical education’ (PE) might be mention only once, then Authors could use the abbreviation (PE) through the whole text. There is a luck of the description of the important factor which goes in line with the engagement and disaffection during the PE process: support of PE  teacher and classmates/peers, even parents. I wander how it would come that fifth and sixth-grade elementary school students are aged 10-14 years? (line 24-25). Please, provide the references for the information which you mentioned in the line 108/09 saying of Mexican population inactivity.

Material & Methods

How many public and how many private school were involved in the study? Is the PE process the same in those institutions (e.g. based on the national curricula of PE)? What of the quality of facilities there, are they similar? (line 118). What is the difference between federal and state schools? Do they implement the same content? (line 120). Lines 130-133 describe Likert scale in my point of view, aren’t they? What Authors meant writing “doctors”? PhD doctors? (l. 161). Wouldn’t be better to write “compared” instead of “contrasted”? (l. 166). Can you proof the final results of the pilot study? (l. 170). All abbreviations should be full described first. (l. 178).

Discussion

The discussion needs to be supplemented in order to strengthen and rewrite it using the references of authors writing about this phenomenon.

Suggestions:

Korcz (et.al.), Is social support during physical education lessons associated with body mass index status, gender and age? South African Journal for Research in Sport, Physical Education and Recreation 2018 : 40 (2) , 53-68 ; 3 ryc., 4 tab., bibliogr. 52 poz. p-ISSN: 0379-9069; http://web.a.ebscohost.com/ehost/pdfviewer/pdfviewer?vid=3&sid=556c9b32-b3ef-4d6a-bdc8-0bdfeb324da4%40sdc-v-sessmgr05

Bronikowski (et.al.), PE Teacher and Classmate Support in Level of Physical Activity : The Role of Sex and BMI Status in Adoledcents from Kosovo, BioMed Research International 2015 : Vol. 2015 (Article ID 290349) , 8 s. ; 5 tab., bibliogr. 36 poz. p-ISSN: 2314-6133, Open Access DOI: 10.1155/2015/290349, http://downloads.hindawi.com/journals/bmri/2015/290349.pdf

Anne Cox , Nicole Duncheon & Lindley McDavid, Peers and Teachers as Sources of Relatedness Perceptions, Motivation, and Affective Responses in Physical Education, Pages 765-773 | Published online: 23 Jan 2013.

Martin Hagger , Nikos L.D. Chatzisarantis , Vello Hein , István Soós , István Karsai , Taru Lintunen, Teacher, peer and parent autonomy support in physical education and leisure-time physical activity: A trans-contextual model of motivation in four nations, Pages 689-711 | Received 01 May 2007, Accepted 28 Jan 2008, Published online: 01 Jul 2009, https://doi.org/10.1080/08870440801956192

Author Response

The authors appreciate the comments and suggestions of the reviewers, which have significantly increased the quality of our manuscript. We hope that the changes made meet your expectations.

The paper seems to be interesting, but in regards of Mexican population (locally). Nevertheless as authors claims there is the first study undertaken in Mexico in that context.

Re: thanks for your comments

Introduction

The expression ‘physical education’ (PE) might be mention only once, then Authors could use the abbreviation (PE) through the whole text.

Re: The abbreviation for PE was included throughout the text except in the writing of instrument items

There is a luck of the description of the important factor which goes in line with the engagement and disaffection during the PE process: support of PE  teacher and classmates/peers, even parents.

Re: L.7-9. The authors decided not to include the description of the factors associated with engagement and disaffection, such as the support of the teacher, classmates and parents, among others, because these variables are not the central part of the study and the number of words in the manuscript could be significantly increased. However, we welcome the suggestions of studies, which served to strengthen the section that mentions that these factors are important for the engagement in the text.

I wander how it would come that fifth and sixth-grade elementary school students are aged 10-14 years? (line 24-25).

Re: The Ministry of Public Education of Mexico establishes that only children who are 6 years old until December 31 can enroll in the month of August.

Assume that the school period begins in August 2016, in this case, children who were born from Aug - Dec 2009 and Jan - Jul 2010 should enter. Therefore, children born during August - December 2009 entered primary school with 7 years old and those born from August to December 2010 entered with 6 years.

Therefore, in 1st grade there are children with 6 and 7 years, in 2nd with 7 and 8 years, in 3rd with 8 and 9 years, in 4th with 9 and 10 years, in 5th of 10 and 11 years and in sixth with 11 and 12 years. Children with 13 (n = 12) and 14 (n = 2) years are probably lagging students or entered 7 and 8 years to first grade.

https://www.aefcm.gob.mx/preinscripciones-gobmx/archivos-2020/convocatoria-2020v4.pdf

Please, provide the references for the information which you mentioned in the line 108/09 saying of Mexican population inactivity.

Re: L.108. The reference of the work with which we argued the information was added.

Material & Methods

How many public and how many private school were involved in the study? Is the PE process the same in those institutions (e.g. based on the national curricula of PE)?

Re: L.115-123. We appreciate the observation and apologize for the mistake. In this study, only students from public elementary schools were considered. This information was corrected in the text.

What of the quality of facilities there, are they similar? (line 118).

Re: In general, the quality of facilities in public schools are they similar.

What is the difference between federal and state schools? Do they implement the same content? (line 120).

Re: In this case, the state schools are administered by the government of the state of Nuevo León and the federal schools are administered by the federal government, that is, by the Mexican Republic. The curricula in both is determined by the secretary of public education of Mexico Art. 25, see at https://legalzone.com.mx/wp-content/uploads/2019/10/Nueva-Ley-General-de- Education-LegalzoneMx.pdf

Lines 130-133 describe Likert scale in my point of view, aren’t they?

Re: L.129. Yes, the questions are answered on a Likert scale. This information was added.

What Authors meant writing “doctors”? PhD doctors? (l. 161).

Re: L.160. Thanks for the observation, it was specified in the text that they are PhD.

Wouldn’t be better to write “compared” instead of “contrasted”? (l. 166).

Re: L. 164. Thanks for the observation, the change was made.

Can you proof the final results of the pilot study? (l. 170).

Re: L. 167-171. The results of the pilot study were added.

All abbreviations should be full described first. (l. 178).

Re: All abbreviations were added the first time the terms appear, except in the writing of the items

Discussion

The discussion needs to be supplemented in order to strengthen and rewrite it using the references of authors writing about this phenomenon.

Re: The authors welcome the suggestions of the reviewers; however, the proposed studies do not include the instrument or the variables that measure the instrument used, therefore, we do not consider it convenient to discuss with studies that do not use the same measuring instrument. However, they were included and were very useful to strengthen the introduction. Likewise, the discussion was strengthened and rewritten with other studies found that use the same instrument and measure the same variables.

Reviewer 2 Report

I realize that a great work and time has been devoted to this paper. This is a topic of great significance to emotional wellbeing of students so I appreciate authors examining this topic.

The paper has a lot of strengths but I think that some changes should be recommended.

Abstract:

Please, avoid using abbreviations in the abstract.

I don’t know why the authors don’t choose engagement and disaffection as keywords.

Introduction:

I would suggest to the authors to avoid the word “we/our”, writing in impersonal mode as scientific style.

Is this test (CEDS) validated on Spanish population?

Why specifically in physical activity? Please, specify this in the manuscript.

And why specifically in fifth and sixth-grade elementary school students? So please, specify both.

Methodology:

The authors must specify the type of sampling, the response rates and loss to follow-up.

Results, Discussion and Conclusions:

The results are good and exhaustive.

There is no comparison with other studies and the discussion is severely lacking.

I hope that these recommendations do not discourage the authors and I want to recommend the authors to continue working on this paper.

Author Response

The authors appreciate the comments and suggestions of the reviewers, which have significantly increased the quality of our manuscript. We hope that the changes made meet your expectations

I realize that a great work and time has been devoted to this paper. This is a topic of great significance to emotional wellbeing of students so I appreciate authors examining this topic.

The paper has a lot of strengths but I think that some changes should be recommended.

Re: Thanks for the feedback

Abstract:

Please, avoid using abbreviations in the abstract.

Re: All abbreviations of the abstract were removed

I don’t know why the authors don’t choose engagement and disaffection as keywords.

Re: L.34. We appreciate the observation and the suggested keywords were added

Introduction:

I would suggest to the authors to avoid the word “we/our”, writing in impersonal mode as scientific style.

Re: We appreciate the observation. Changes were made to the text and marked in red

Is this test (CEDS) validated on Spanish population?

Re: We have not found validation studies of (CEDS) with Spanish population

Why specifically in physical activity? Please, specify this in the manuscript.

Re: The reason for conducting the study in the context of physical education is described in lines 90-97

And why specifically in fifth and sixth-grade elementary school students? So please, specify both.

Re: L.121-123. The following was added: Fifth and sixth grade students were chosen because children who belong to the final years of third childhood and early adolescence are at the highest level of cognitive development and will not have any complications when responding to the instruments

Methodology:

The authors must specify the type of sampling, the response rates and loss to follow-up.

Re: The type of sampling is specified in lines 120 and 121.

L.174: We add the follow: Missing data rate were very small (0,14%) that it was not considered necessary impute the data.

Results, Discussion and Conclusions:

The results are good and exhaustive.

There is no comparison with other studies and the discussion is severely lacking.

Re: The discussion was extended and improved with other studies; however, this was somewhat complicated due to the lack of studies that used the instrument used in the present study

I hope that these recommendations do not discourage the authors and I want to recommend the authors to continue working on this paper.

Re: Thank you very much for your recommendations and words of encouragement
